# Association of Acupuncture Treatment with Mortality of Type 2 Diabetes in China: Evidence of a Real-World Study

**DOI:** 10.3390/ijerph17217801

**Published:** 2020-10-25

**Authors:** Mengyun Sui, Long Xue, Xiaohua Ying

**Affiliations:** NHC Key Laboratory of Health Technology Assessment, School of Public Health, Fudan University, Dongan Road 130, Shanghai 200032, China; 18111020027@fudan.edu.cn (M.S.); 18111020030@fudan.edu.cn (L.X.)

**Keywords:** diabetes mellitus, acupuncture, mortality, traditional Chinese medicine

## Abstract

The prevalence and mortality rates of diabetes are increasing globally, posing severe challenges to health systems. Acupuncture is used worldwide as a non-drug treatment for diabetes. However, empirical evidence of the effect of combined acupuncture and drug treatments on diabetic-associated mortality is limited. This study aimed to examine this association of acupuncture treatment with mortality of type 2 diabetes based on real-world data. A four-year cohort study was conducted in Shanghai between 2015 and 2018, The database consisted of 37,718 patients (acupuncture group: 6865 type 2 diabetes mellitus (T2DM) patients, non-acupuncture (control) group: 30,853 T2DM patients) in 2016. The objective was to analyze the impact of receiving acupuncture prescriptions for diabetes in 2016 on all-cause mortality in 2018 based on real-world data. An Inverse Probability Weighted Regression Adjustment (IPWRA) and Propensity Score Matching (PSM) were used to minimize the bias due to potential confounding variables to increase the reliability of differences in comparisons between the two groups. Our inverse probability weighted regression results suggest that the coefficient of the key dependent variable of accepted acupuncture in 2016 was negative (coefficient: −0.0002; 95% CI: −0.0024–0.0019, *p* = 0.857), but it is not statistically significant. In robustness check, PSM with the nearest-neighbor method with replacement at a 1:4 ratio and 1:3 ratio and kernel matching showed that the average treatment effect was negative. Therefore, there was a negative correlation between acupuncture combined with other drugs and the mortality of diabetic patients, but it was not statistically significant.

## 1. Introduction

### 1.1. Prevalence of Diabetes

Diabetes mellitus has shown an alarming increase, becoming a global epidemic [1]. According to the most recent International Diabetes Federation (IDF) Diabetes Atlas 2019, the prevalence of diabetes worldwide tripled from 151 million in 2000 to 463 million in 2019 and is projected to reach 700 million by 2045. Diabetes is expected to be present in 10.9% of the global population by 2045 [2]. Patients with diabetes are prone to microvascular complications (nephropathy, retinopathy, and neuropathy) and macrovascular complications (concomitant atherosclerosis and coronary artery disease) [3]. The number of deaths resulting from diabetes and its complications was estimated to be 4.2 million in 2019. The ultimate value of diabetes prevention lies in reduced blood sugar, incidence of complications, and mortality [4].

### 1.2. Introduction of Acupuncture

Acupuncture therapy is a significant part of traditional Chinese medicine (TCM). There are various types of acupuncture, including manual acupuncture, electroacupuncture, ear acupuncture, moxibustion, and acupoint injection [5]. It involves the treatment of disease by inserting needles along specific pathways or meridians, according to the disease being treated. Acupuncture is widely practiced globally, including in countries such as Japan, Taiwan, China, and the United States [6,7,8]. Many patients use a combination of conventional medicine and acupuncture to treat diabetes [9]. Acupuncture has unique advantages. For example, its approach is holistic in that it hinges on the belief that the human body is an organic whole, which is mainly connected by the meridian system [10].

### 1.3. Strengthening Acupuncture in China

The World Health Organization (WHO) recommends that countries take effective measures to prevent and treat diabetes [11]. Accordingly, the WHO and the National Institutes of Health reported that acupuncture could be an effective means to treat diabetes [12]. As acupuncture is indigenous to China, the Chinese government is paying more attention to its development and promotion. In the Chinese Guidelines for the Prevention and Treatment of type 2 diabetes mellitus (T2DM) (2017 edition), TCM treatment has been included. China is increasing the coverage of acupuncture in community health service centers and family doctor teams. Meanwhile, with the internationalization of TCM, it is necessary to prove the effectiveness of this practice.

To meet the demand for acupuncture in diabetes treatment, top-down policies from the Chinese government have been promulgated [13]. For example, in 2009, the State Administration of TCM issued an implementation plan for appropriate TCM technology [14]. In 2016, the State Administration of TCM announced a project that promotes the integration of acupuncture into the regular diabetic care provided by the family physician [15]. Since then, an increasing number of patients with diabetes have been using acupuncture as an alternative medical therapy.

Shanghai is a pioneer city for the family doctor system and TCM services. The family doctor team usually has one or two core general practitioners and is equipped with a TCM physician, public health physician, nurse, and physician assistant [16]. The government is committed to providing acupuncture knowledge and skills training to family doctors, thereby increasing the number of TCM doctors and expanding the integration of acupuncture into family doctor teams. The family doctor provides Western medicine, TCM, acupuncture and moxibustion and other related services. In 2018, a total of 6.66 million family doctors in Shanghai had signed contracts with 30% of permanent residents; moreover, more than 84% of patients with diabetes had signed contracts [17]. Acupuncture is frequently applied in conjunction with Western medicine to relieve symptoms in patients with diabetes.

### 1.4. The Purpose of the Study

Previous studies have reported the relationship between acupuncture and blood glucose levels and complications in patients with diabetes [18,19]. However, there is insufficient evidence to support these results because of the small sample size, uncertain outcomes, and a lack of real-world situations. Although many people enjoy using acupuncture, the mainstream scientific approach based on evidence and outcome reproduction cannot fully demonstrate the value of acupuncture, and the evidence of the contribution of acupuncture to health is limited.

To bridge the gap between acupuncture therapy and its effect on type 2 diabetes mellitus (T2DM), the participants of this study were categorized into two groups—control and acupuncture groups. Only one goal has been considered in this study. The objective was to analyze the impact of received acupuncture for diabetes in 2016 on all-cause mortality in 2018 based on real-world data. We used baseline characteristics of patients in 2016 to estimate diabetes death in 2018. We explored the long-term health effects of receiving acupuncture prescriptions. The hypothesis of this study was compared with that of the control group; the acupuncture group had a lower mortality rate. The mortality rate was the primary outcome of the research.

## 2. Materials and Methods

### 2.1. Study Design

In this study, we utilized a three-stage sampling design to select T2DM patients (stratified by the community health center, family physician team, and T2DM patients). First, out of the 240 community health service centers in the 16 administrative regions of Shanghai, one-third were included in the study. These 80 centers were selected through stratified sampling of the administrative divisions. Then, three family physician teams were drawn from each center, and a total of 240 teams were included. Finally, 150 patients with T2DM who received TCM were chosen from the family physician team by simple random sampling. We defined the treatment group as those who received acupuncture therapy. The study population selection is shown in Figure 1.

### 2.2. Data Source and Population Selection

This study used panel data deposited in the Health Information Center Database of Shanghai, from patients who received TCM from 1 January 2015 to 31 December 2018. Quarterly records of patients over these four years were collected, but we ended up using the year data as the unit of analysis. The datasets consisted of demographic characteristics, outpatient and inpatient health service utilization of different hospital types (primary, secondary, tertiary hospital), number of acupuncture prescriptions, cost of different hospital types, death, and information on complications. We defined disease according to the International Classification of Diseases, 10th Revision (ICD-10). The database consisted of 37,718 patients (acupuncture group: 6865 T2DM patients, non-acupuncture (control) group: 30,853 T2DM patients) in 2016.

### 2.3. Research Variables

#### 2.3.1. Dependent Variable

We included one critical outcome—i.e., occurrence of mortality, as a dependent variable. Mortality is a binary variable, and it was recognized when the patient died in 2018.

#### 2.3.2. Independent Variable

The key variable was the choice of which group should receive acupuncture—defined as “1” for acupuncture and “0” for the control group. Categorial variables consisted of patients’ gender, age, other chronic diseases, treatment cost, and residential location. Gender was categorized as male and female; four age ranges were stratified (<60, 60–69, 70–79, and ≥80 years); four levels of other chronic diseases were categorized (0, 1, 2 and ≥3 chronic diseases); two classes for the residential location were categorized (urban and outskirts); the cost per year was divided into four groups (≤2000, 2001–7000, 7001–15,000 and ≥15,001 yuan). We collected information on other chronic diseases, such as hypertension, coronary heart disease, stroke, arrhythmia, and angina. The complication variable was binary, with “yes” and “no” as options.

The regression formula is as follows:$$Y_{i}=\beta_{0}+\beta_{1}Treat+\beta X_{i}+\varepsilon_{i}$$
where *Y_i_* refers to diabetic *i* who died or not; treat denotes whether they received acupuncture treatment; *X_i_* indicates covariate variables, including gender, cost, age, residential location and other chronic disease. ε_i_ is the error term.

### 2.4. Statistical Analysis

Descriptive statistics were used to demonstrate the baseline characteristics of the patients with T2DM. Percentages were used to analyze categorical data, and means were adopted to analyze continuous data.

In this study, we used baseline data for diabetes in 2016 and death data for 2018 to estimate the effect. Due to the retrospective cohort design of our study and the possibility of selection bias, we considered that the demographic characteristics and disease status of the patients could affect the results. Therefore, an Inverse Probability Weighted Regression Adjustment (IPWRA) and Propensity Score Matching (PSM) were used to minimize the bias due to potential confounding variables to increase the reliability of differences in comparisons between the two groups [16,17]. We used IPWRA to estimate the relationship of whether received outpatient visits of traditional Chinese Medicine for diabetes in 2016 with all-cause mortality in 2018. IPWRA is one approach to estimate unbiased treatment effects when we have confounding factors. Firstly, the IPWRA method generated IP (Inverse Probability) weights according to individual’s values of treatment and covariate and assigns the inverse of probability of treatment for treated individuals and the inverse probability of not being treated for control individuals [20,21]. Then, it re-estimates the outcome model using these new weights. In this study, we use gender, age, residential location, cost and other chronic disease to estimate IP weights, then new weights and a logistic regression model were used to estimate outcomes.

In a robustness check, we used three methods to estimate the Average Treatment Effect (ATE)—PSM of a 1:4 ratio, 1:3 ratio and kernel matching, respectively. We used baseline data from patients with diabetes in 2016 for matching. Patients’gender, age, other chronic, cost, residential location are matching variables. Propensity score, logistic regression on patients’ demographics information for each eligible subject was matched according to the score by using the nearest-neighbor method at a 1:4 ratio with 0.02 in the caliper [22]. The 50 times bootstrap method was employed to obtain the robust standard error.

For all data sets, differences were considered statistically significant when the *p* value < 0.05. All analyses were performed using STATA software (version 16.0).

## 3. Results

### 3.1. Baseline T2DM Patient Characteristics

Table 1 describes the baseline patient characteristics of the treatment and control groups. Both the acupuncture and control groups had a higher proportion of females. The average age of the acupuncture group was two years older than that of the control group. A higher proportion of patients in the acupuncture group had other chronic diseases. The results indicated that people with T2DM and other chronic conditions are more likely to seek non-drug treatment. Overall, acupuncture group patients were more likely to be older, female, have other chronic disease. The outpatient and inpatient visits in the treatment group were higher than those in the control group. In the treatment group, mean prescriptions of acupuncture were 3.26 ± 4.74 per year, and medical expenses in the same period were approximately 1.6 times that of the control group.

### 3.2. Inverse Probability Weighted Regression Results

In 2018, there was 45 diabetes patients in the acupuncture group who died—a mortality of 0.67%; in the control group there were 186 diabetes patients who died—a mortality of 0.61%. After controlling for confounders, the results were collected and are shown in Table 2. Our inverse probability weighted regression results suggest that the coefficient of the key dependent variable of accepted acupuncture in 2016 was negative (coefficient: −0.0002; 95% CI: −0.0024–0.0019, *p* = 0.857), but it is not statistically significant. Compared with women, being a man is a risk factor for death in diabetes. Age is also a dangerous factor for diabetes, and the older the person with diabetes, the greater the probability of death. An age over 70 years old increases the threat to diabetic patients significantly (coefficient: 0.0056; 95% CI: 0.0030–0.0082, *p* < 0.001).

### 3.3. Robust Test Results

After using the PSM method at 1:4 ratio, 970 patients were assigned to the acupuncture group and successfully matched with 5823 patients in the control group. After matching, the mortality rate in the acupuncture group was 0.74% and that in the control group was 1.02%. While many differences were statistically significant before the PSM procedure (*p* < 0.05, Table 3), there were few substantively important differences between the acupuncture group and the control group after adjustment (*p* > 0.05). As shown in Table 4, the average treatment effect in the PSM of 1:4 method was negative but not statistically significant (coefficient: −0.0014, 95% CI: −0.0072, 0.0042, *p* = 0.611). Using the PSM method at 1:3 ratio, the coefficient is also negative (coefficient: −0.0034, 95% CI: −0.0105, 0.0036, *p* = 0.344). In the Kernel Matching, the coefficient is −0.0002.

## 4. Discussion

This nationwide study analyzed the impact of receiving acupuncture for diabetes in 2016 on all-cause mortality in 2018 based on real-world data from mainland China. The results suggest a negative correlation between acupuncture combined with other drugs and the mortality of diabetes, but it was not statistically significant. The mortality rate was lower in the acupuncture combined with other drug groups than in the control group, but the difference was not significant. To the authors’ knowledge, this study is one of the few to suggest that acupuncture reduces T2DM mortality using real-world data. There are reasons why there is a negative relationship between acupuncture treatment and mortality among patients with T2DM. First, acupuncture can improve insulin function, glucose metabolism, and glucose-related hormone levels by stimulating acupoints, thus achieving the goal of lowering blood glucose. At the same time, acupuncture can stimulate the biological activity of mitochondrial respiratory chain-related enzymes, improve mitochondrial function, and thus enhance the sensitivity of islet cells. These related mechanisms provide theoretical support for the acupuncture treatment of diabetes [5,23]. Western medicine can reduce the blood sugar levels of patients with diabetes and the incidence of complications, but it is not possible to achieve a complete cure. Acupuncture, in combination with Western medicine, has better efficacy and fewer side effects than single medicine alone, consequently reducing damage to the body caused by drugs administered to diabetic patients.

Our conclusion is that a negative correlation between acupuncture combined with other drugs and the mortality of diabetic, to some extent, can support the previous literature, comprising studies based on diabetic model rats, clinical trials, and meta-analyses. For example, some studies have clarified the mechanism and potential advantages of acupuncture in improving diabetic patients’ blood sugar and its complications through animal experiments [24]. The compound laser acupuncture–moxibustion had positive effects on the regulation of hyperglycemia and insulin resistance in T2DM rats [25,26]. In terms of clinical indicators, a study involving 1943 people with diabetes showed that acupuncture baseline treatments reduced fasting blood sugar (FBS), 2-h blood glucose, and glycated hemoglobin levels [18]. Participants in the acupuncture group of another study were needled at CV-12 and showed a significant drop in random blood glucose levels [19]. Yet another study showed that a combination of metformin and acupuncture improves the body mass index; body weight; fasting insulin, FBS, triglyceride, low-density lipoprotein cholesterol, and high-density lipoprotein cholesterol levels [27]. In terms of treating complications, the findings of the acupuncture trial may establish the value of acupuncture therapy for the improvement of peripheral nerve function and subjective perception relative to diabetic peripheral neuropathy [28,29,30,31]. Acupuncture treatment is also a cost-effective option. Acupuncture therapy improves vascular circulation and wound healing as an alternative therapy for the treatment of diabetic foot [32].

Our study has three advantages. First, based on the perspective of evidence-based medicine, combined with real-world data, we answered one key question. We chose mortality as the outcome indicator that could reflect the relationship between real-world acupuncture treatment and the mortality rate in patients with T2DM. At present, most articles have utilized process indicators. Second, our study sample came from a credible source (Shanghai Health Information Statistics Center) and was sufficiently large, had good representativeness, and less missing data. Third, confounding factors were eliminated by the IPWRA and PSM. IPWRA and PSM are popular econometric methods at present. Nonetheless, our study has a few limitations. First, although we used IPWRA and PSM to eliminate the differences between the two groups, we might not have eliminated potential confounding factors entirely, including patients′ lifestyle habits such as smoking, drinking, and exercise. Second, the conclusion drawn by IPWRA and PSM is not a causal relationship, and thus further research is required. Third, our data represent a retrospective cohort, and the study has not yet been prospectively designed.

## 5. Conclusions

We have demonstrated that the association of acupuncture combined with other-drug groups with mortality is negative but statistical insignificance. Further studies with a larger sample of patients with T2DM and using econometric methods combined with the current popular big data machine learning methods would validate the impact of acupuncture on the mortality rate in patients with T2DM and provide objective evidence for acupuncture treatment of diabetes.

## Figures and Tables

**Figure 1 ijerph-17-07801-f001:**
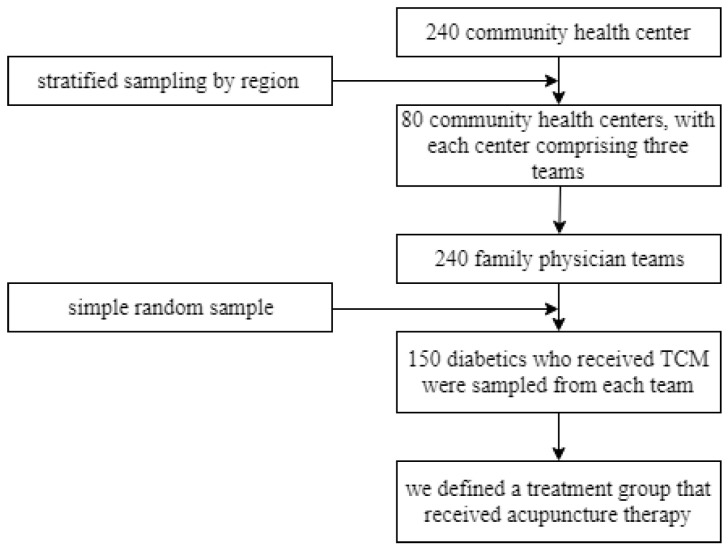
The sampling protocol of this study. TCM: traditional Chinese medicine.

**Table 1 ijerph-17-07801-t001:** Baseline characteristics of type 2 diabetes mellitus (T2DM) patients in 2016.

Characteristics	Acupuncture Group (N = 6865)			Non-Acupuncture Group (N = 30,853)		
	Mean	SE	95%CI	Mean	SE	95% CI
Gender (%)						
Male	34.64	0.006	33.43–35.87	40.98	0.002	40.44–41.53
Female	65.35	0.006	64.13–66.57	59.01	0.002	58.46–59.56
Age (years)						
Mean	70.04	0.125	69.79–70.28	67.91	0.057	67.79–68.02
<60	9.96	0.003	9.27–10.69	19.29	0.002	18.85–19.73
60–69	33.38	0.006	32.28–34.51	38.79	0.003	38.25–39.34
70–79	25.53	0.005	24.51–26.58	27.09	0.003	26.60–27.59
≥80	31.11	0.006	30.03–32.22	14.81	0.002	14.42–15.21
Residential location (%)						
Urban	44.22	0.006	42.95–45.49	34.93	0.002	34.39–35.46
Suburban	55.77	0.006	54.50–57.04	65.07	0.002	64.53–65.60
Chronic (%)						
0	9.77	0.003	9.09–10.49	20.18	0.002	19.73–20.63
1	19.16	0.004	18.25–20.11	29.91	0.002	29.40–30.42
2	23.10	0.005	22.12–24.11	23.92	0.002	23.44–24.40
≥3	47.95	0.006	46.77–49.13	25.99	0.002	25.50–26.48
Total cost (yuan)						
Mean	16,571.99	177.64	16,223.74–16,920.24	10,098.17	61.94	9976.74–10,219.60
≤2000	5.07	0.002	4.57–5.61	20.32	0.002	19.87–20.77
2001–7000	16.35	0.004	15.50–17.25	31.53	0.003	31.01–32.05
7001–15,000	29.30	0.005	28.24–30.39	27.76	0.003	27.26–28.26
≥15,001	49.26	0.006	48.08–50.44	20.39	0.002	19.94–20.84
Outpatient visits						
Number of primary-level hospitals	39.51	0.355	38.81–40.21	26.63	0.135	26.36–26.90
Number of secondary-level hospitals	7.43	0.138	7.16–7.71	5.35	0.052	5.25–5.45
Number of tertiary-level hospitals	8.82	0.144	8.53–9.10	6.50	0.056	6.39–6.61
Inpatient visits						
Number of primary-level hospitals	0.012	0.002	0.01–0.02	0.012	0.000	0.01–0.01
Number of secondary-level hospitals	0.278	0.012	0.25–0.30	0.145	0.003	0.13–0.15
Number of tertiary-level hospitals	0.271	0.010	0.25–0.29	0.149	0.003	0.14–1.56
Total prescriptions of acupuncture	3.26	0.062	3.14–3.38	0.00	0.000	0.00

**Table 2 ijerph-17-07801-t002:** Inverse probability weighted regression results.

Death in 2018	Coefficient	95%CI	SE	*p*-Value
Treat 2016	−0.0002	−0.0024–0.0019	0.0011	0.857
Gender (%)				
Male	0.0040 ***	0.0016–0.0064	0.0012	0.001
Age (%)				
60–69	0.0016	−0.0005–0.0038	0.0011	0.140
70–79	0.0056 ***	0.0030–0.0082	0.0013	0.000
≥80	0.0192 ***	0.0144–0.0239	0.0024	0.000
Chronic (%)				
1	0.0004	0.0032–0.0024	0.0014	0.774
2	0.0003	0.0042–0.0035	0.0019	0.850
≥3	0.0025	0.0061–0.0008	0.0017	0.145
Residential location (%)				
Urban	−0.0024 ***	−0.0046–−0.0003	0.0011	0.027
Cost (%)				
2001–7000	0.0050 ***	0.0022–0.0007	0.0014	0.000
7001–15,000	0.0049 ***	0.0019–0.0079	0.0015	0.001
≥15,001	0.0063 ***	0.0029–0.0095	0.0016	0.000
Constant	−0.0026	−0.0055–0.0002	0.0014	0.075

*** *p* < 0.01, CI: Confidence Interval, SE: Standard Error.

**Table 3 ijerph-17-07801-t003:** Comparable characteristics between the two groups before and after matching.

Characteristics	Before Matching			After Matching		
	Acupuncture Group	Non-Acupuncture Group	*p* Value	Acupuncture Group	Non-Acupuncture Group	*p* Value
Gender (%)						
Male	34.57	40.97	<0.001	34.57	34.57	1.000
Age (%)						
60–69	39.15	38.84	0.647	39.15	39.15	1.000
70–79	29.95	27.11	<0.001	29.95	29.95	1.000
≥80	19.25	14.78	<0.001	19.25	19.25	1.000
Chronic (%)						
1	22.49	29.92	<0.001	22.49	22.49	1.000
2	27.07	24.01	<0.001	27.07	27.07	0.979
≥3	39.12	26.10	<0.001	39.12	39.12	0.981
Cost (%)						
2001–7000	19.13	31.61	<0.001	19.13	19.13	1.000
7001–15,000	34.39	27.88	<0.001	34.39	34.39	1.000
≥15,001	40.62	20.48	<0.001	0.62	0.62	1.000
Residential location (%)						
Urban	44.22	34.88	<0.001	44.22	44.22	1.000

**Table 4 ijerph-17-07801-t004:** The average treatment effect of the three methods.

Method	Coefficient	95%CI	Standard Error	Z	*p*
1:4 PSM (bootstrap 50 times)	−0.0014	−0.0072, 0.0042	0.0029	−0.51	0.611
1:3 PSM (bootstrap 50 times)	−0.0034	−0.0105, 0.0036	0.0036	−0.95	0.344
Kernel Matching	−0.0002	−0.0022, −0.0016	0.0009	−0.30	0.766

PSM: Propensity Score Matching.

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
