# Peer review of "Association of Acupuncture Treatment with Mortality of Type 2 Diabetes in China: Evidence of a Real-World Study"

_ijerph, 2020, doi:10.3390/ijerph17217801_

Round 1
Reviewer 1 Report
Abstract: At the end the discussion that the 'treatment frequency did not increase the protectiveness significantly' seems to contradict the final statement that the 'frequent use of acupuncture may be a protective factor.' This will need rewording to clarify the meaning.
Section 1.4: In would be helpfulto insert a reference or two after the first sentence.
Section 2.4 and Figure 2: 'Propensity score matching' needs to be much more clearly described, as the results of the study depend critically on this. Section 3.2 and Table 2 describe the results of the matching but not the detail of the process. This is actually quite critical.
Table 1: Poorly laid out and difficult to read.
Discussion: The references provided as to the putative mechansisms of acupuncture in diabetes are not evaluated for their rigor. This should probably be done - even with just a statement or two to provide some context.
Overall this is a very important paper, if the propensity matching is a robust process.
Author Response
Thank you for your suggestion.

Reviewer 2 Report
It is interesting that authors conducted a four-year cohort study to examine the association of acupuncture treatment with mortality of T2DM in china, I think several concerns listed below will need to be addressed to improve paper quality
- I think the second paragraph (The World Health Organization …. the effectiveness of this practice) in discussion belong to introduction part.
- I think the third paragraph in discussion part is not appropriate. Three main reasons that authors presented are only postulation. I think authors should provide the evidence for each reason.
- I think the sentence (“ The previous literature, comprising studies based on diabetic model rats, clinical trials, and meta-analyses, supports our conclusion.”) in the fourth paragraph in discussion part is not appropriate. The conclusion should be supported by the result of this study. The conclusion of this study is that the mortality rate is lower in patients with T2DM in the acupuncture combined with other drugs group than in the control group, not the reasons why acupuncture is a protective factor for mortality among patients with T2DM.
- I think the sentence (“More frequent use of acupuncture may be a protective factor for mortality in T2DM. “) in the conclusion is not appropriate. The results of your study is that
the use of acupuncture treatment twice was a significant protective factor for diabetes (OR=0.489, 95% CI=0.259–0.921, P=0.027). Increasing the treatment frequency to 3, 4-6, and >6 times did not increase the protectiveness significantly. The conclusion should be written based on the results of the study.
Author Response
Thank you for your suggestion.
